# Construction and Validation of Mortality Risk Nomograph Model for Severe/Critical Patients with COVID-19

**DOI:** 10.3390/diagnostics12102562

**Published:** 2022-10-21

**Authors:** Li Cheng, Wen-Hui Bai, Jing-Jing Yang, Peng Chou, Wan-Shan Ning, Qiang Cai, Chen-Liang Zhou

**Affiliations:** 1Department of Critical Care Medicine, Eastern Campus, Renmin Hospital of Wuhan University, Wuhan 430200, China; 2Department of Hepatobiliary Surgery, Eastern Campus, Renmin Hospital of Wuhan University, Wuhan 430200, China; 3Department of Vascular Surgery, North Campus of Shanghai Ninth People’s Hospital, Shanghai Jiao Tong University School of Medicine, Shanghai 201900, China; 4Department of Clinical Laboratory, Union Hospital, Tongji Medical College, Huazhong University of Science and Technology, Wuhan 430022, China; 5Department of Neurosurgery, Renmin Hospital of Wuhan University, Wuhan 430200, China

**Keywords:** COVID-19, death, nomogram, prognosis

## Abstract

**Objective:** A nomograph model of mortality risk for patients with coronavirus disease 2019 (COVID-19) was established and validated. **Methods:** We collected the clinical medical records of patients with severe/critical COVID-19 admitted to the eastern campus of Renmin Hospital of Wuhan University from January 2020 to May 2020 and to the north campus of Shanghai Ninth People’s Hospital, Shanghai JiaoTong University School of Medicine, from April 2022 to June 2022. We assigned 254 patients to the former group, which served as the training set, and 113 patients were assigned to the latter group, which served as the validation set. The least absolute shrinkage and selection operator (LASSO) and multivariable logistic regression were used to select the variables and build the mortality risk prediction model. **Results:** The nomogram model was constructed with four risk factors for patient mortality following severe/critical COVID-19 (≥3 basic diseases, APACHE II score, urea nitrogen (Urea), and lactic acid (Lac)) and two protective factors (percentage of lymphocyte (L%) and neutrophil-to-platelets ratio (NPR)). The area under the curve (AUC) of the training set was 0.880 (95% confidence interval (95%CI), 0.837~0.923) and the AUC of the validation set was 0.814 (95%CI, 0.705~0.923). The decision curve analysis (DCA) showed that the nomogram model had high clinical value. **Conclusion:** The nomogram model for predicting the death risk of patients with severe/critical COVID-19 showed good prediction performance, and may be helpful in making appropriate clinical decisions for high-risk patients.

## 1. Introduction

COVID-19, caused by severe acute respiratory syndrome coronavirus 2 (SARS-CoV-2), has been defined by the World Health Organization (WHO) as a “public health emergency of international concern”, seriously threatening the safety of human life [1]. Studies have shown that the 28 d mortality rate of patients with severe COVID-19 has been as high as 61.5% [2]. Therefore, early identification of critical patients and early accurate treatment can effectively reduce COVID-19-related mortality rates [3]. In recent years, many risk models for predicting the death of patients with COVID-19 have been published internationally [4,5,6,7,8], but most of them had only internal validation, without external validation across regions. In March 2022, an epidemic of the Omicron mutant broke out in Shanghai. On 17 April 2022, the intensive-care team for Shanghai aid from Remmin Hospital of Wuhan University entered the ICU ward of the north campus of Shanghai Ninth People’s Hospital, Shanghai JiaoTong University School of Medicine. In the course of treatment, we learned from the valuable experience of anti-epidemic approaches taken in Wuhan and summarized the characteristics of the anti-epidemic approach taken in Shanghai. Based on the clinical analysis of common test results and the universality of detection in patients, we constructed a nomogram model of mortality risk using the data of severe/critical COVID-19 patients in Wuhan, and externally verified the data of severe/critical COVID-19 patients in Shanghai.

## 2. Materials and Methods

### 2.1. General Data

This study was a double-center, retrospective, and observational study. It was approved by the ethics committee of Renmin Hospital of Wuhan University and Shanghai Ninth People’s Hospital, affiliated with Shanghai JiaoTong University School of Medicine. The study obtained ethics exemption. We collected the clinical medical records of patients with severe/critical COVID-19 admitted to the eastern campus of Renmin Hospital of Wuhan University from January 2020 to May 2020 and those admitted to the north campus of Shanghai Ninth People’s Hospital, Shanghai JiaoTong University School of Medicine from April 2022 to June 2022. The data of the patients admitted to the former hospital were used as the training set (*n* = 254), and data of the patients in the latter were used as the validation set (*n* = 113). The patients with severe/critical COVID-19 were taken as the starting point and the patients who died in hospital or were followed-up for 28 days were taken as the end point. The follow-up date was 26 June 2022. 

#### 2.1.1. Inclusion Criteria

The inclusion criteria for this study were as follows: (1) age ≥ 18 years old; (2) patients with complete case data; (3) positive for the novel coronavirus nucleic acid detected by fluorescence quantitative PCR.

##### Enrollment Criteria for Patients in Wuhan

According to recommendations in the first edition of the COVID-19 diagnosis and treatment guidelines [9], those who met one of the following criteria were diagnosed as severe cases: (1) respiratory distress—respiratory rate ≥30 beats/min (RR ≥ 30 bpm); (2) pulse oxygen saturation (SpO2) ≤ 93% on room air at rest state or arterial partial pressure of oxygen (PaO2)/Fraction of inspiration O2 (FiO2) ≤300 mmHg (1 mmHg = 0.133 kPa); (3) pulmonary imaging showing multi-lobar lesions or lesion progression in >50% within 24~48 h; (4) quick sequential organ failure assessment (qSOFA) score ≥1, or combined with pneumothorax; (5) other clinical conditions requiring hospitalization. Those who met one of the following criteria were diagnosed as critically ill cases: (1) respiratory failure; (2) septic shock; (3) combined with other organ failure.

##### Enrollment Criteria for Patients in Shanghai

According to recommendations in the ninth edition of the COVID-19 diagnosis and treatment guidelines [10], those who met one of the following criteria were diagnosed as severe cases: (1) shortness of breath—RR ≥ 30 bpm; (2) SpO2 ≤ 93% on room air at rest state; (3) PaO2/FiO2 ≤ 300 mmHg; (4) progressive aggravation of clinical symptoms, and with > 50% lesions progression within 24~48 h in pulmonary imaging. Those who met one of the following criteria were diagnosed as critically ill cases: (1) respiratory failure occurred and mechanical ventilation was required; (2) shock occurred; (3) complications with other organ failure that require monitoring and treatment.

#### 2.1.2. Exclusion Criteria

The exclusion criteria for this study were as follows: (1) age < 18 years old; (2) pregnant or lactating; (3) treatment period less than 3 days; (4) patients and their families who requested cessation of active treatment and could not receive routine comprehensive treatment; (5) patients who lost contact during follow-up.

### 2.2. Data Collection and Grouping

The general information, physical examination, and auxiliary examination data for the patients were collected, and the classification was determined according to the recommendations of the guidelines [9,10]. The worst blood samples of patients with severe/critical COVID-19 within 24 h after admission were collected, including blood routine, blood biochemistry, coagulation function, blood gas analysis, and other related indexes, and the scores of acute physiology and chronic health evaluation II (APACHE) were calculated.

### 2.3. Statistical methods

SPSS 26.0 and R 4.1.3 were used for statistical analysis and mapping. The measurement data of this study did not obey the normal distribution, and are represented by the median (quartile) (M (Q_L_, Q_U_)). Comparison between groups is expressed by Mann–Whitney U test. The count data are expressed as [*n* (%)], and the Chi-square test, the Chi-square test of continuous correction, or Fisher’s precision probability test were used for the comparison between groups. Whether or not death occurred within 28 days after the diagnosis of severe/critical COVID-19 (secondary outcome index) was taken as the dependent variable. Lasso regression was used to screen the independent variables, and multivariate logistic regression analysis was used to screen the risk factors. The nomogram prediction model was constructed by using R software rms package, and the ability of the model to the predict a prognosis of death was evaluated by ROC. The bootstrap method was used to repeatedly sample 1000 times for internal validation, and the difference of C-index was compared. Furthermore, a calibration curve and DCA were used to evaluate the prediction model. The difference was statistically significant (*p* < 0.05).

## 3. Results

### 3.1. Analysis of Baseline Data of Patients

Of the 254 patients in the training set, 74 (29.13%) eventually died; among these patients, 57 (22.44%) were still COVID-19-positive at the time of death, but 17 (6.69%) patients were COVID-19-negative according to two consecutive coronavirus nucleic acid tests before death (sampling time at least 24 h apart). Of the 113 patients in the validation set, 21 (18.58%) eventually died; among these patients, 5 (4.42%) were still COVID-19-positive according to the novel coronavirus nucleic acid test at the time of death, and 16 (14.16%) patients were COVID-19-negative according to two consecutive novel coronavirus nucleic acid tests before death (at least 24 h apart at the time of sampling). The causes of death were other irreversible serious complications or basic diseases. The baseline data are shown in Table 1. In the training set and the validation set, the patients in the death group were older, more critically ill, had more than three basic diseases, and had higher APACHE II scores (*p* < 0.05). As shown in Table 2, leukocyte count, the neutrophil-to-lymphocyte ratio (NLR), NPR, procalcitonin, direct bilirubin, Lac, B-type natriuretic peptide, troponin, and D-dimer were higher in the group of deceased patients in the training set; however, hemoglobin and albumin were lower in the validation set (*p* < 0.05).

### 3.2. Lasso and Logistic Regression Analysis

A total of 51 potential death-related risk factors were included in the study. The variables of the training set were dimensionally reduced by lasso regression, and the most representative characteristic variables were selected. When selecting the optimal lambda parameters, 5-fold cross-validation was used, and the lambda value with the smallest cross-validation error was taken as the optimal value of the model (Figure 1 and Figure 2), and the number of variables at this time was counted. The results of Lasso regression analysis showed that 11 independent variables, including age ≥65 years, ≥3 basic diseases, APACHE II score, L%, Urea, D-dimer, Lac, NLR, NPR, leukocyte count, and troponin, were the characteristic variables affecting the death of patients with COVID−19. Taking the occurrence of death as the dependent variable and the 11 characteristic variables screened by lasso regression as the independent variables, the results of multivariate logistic regression analysis (Table 3) show that ≥3 basic diseases, APACHE II score, Urea, and Lac were the independent risk factors for death, while L% and NPR were protective factors against death.

### 3.3. Establishment of Nomogram Model

Multivariate logistic regression analysis finally selected 6 independent variables, including ≥3 basic diseases, APACHE II score, Urea, Lac, L%, and NPR to construct the nomogram model, which could be obtained by visual analysis of R language (Figure 3). Critical care physicians could assess the death risk of severe/critical COVID-19 patients in a visual and individualized quantitative way according to these six easily available indicators. The C-index of the training set was 0.880 (95%CI, 0.837~0.923); The larger the C-index was, the better the discrimination of the model was, suggesting the nomogram had better prediction accuracy.

### 3.4. Internal and External Validation of the Model

The AUC of the training set was 0.880 (95% *CI*, 0.837~0.923), with a sensitivity of 87.2% and specificity of 71.6%; the AUC of the validation set was 0.817 (95% *CI*, 0.712~0.923), with a sensitivity of 78.5% and a specificity of 81.0%, indicating good prediction performance, as shown in Figure 4. The training set is internally validated by bootstrap to have the same C-index, that is, the predicted results of the model are consistent with the actual results. The calibration curve was drawn to evaluate the judgment ability of the nomogram model, and Hosmer–Lemeshow goodness-of-fit test was performed. The calibration curve of the training set and the validation set showed that the nomogram and the reference line had goodness of fit, with *p* values of 0.800 and 0.533, respectively (both >0.05), as shown in Figure 5. The Brier scores in the training set and the validation set were 0.049 and 0.086, respectively, which were both close to 0, indicating that the nomogram model predicted a good consistency between the probability of death among severe/critical COVID-19 patients and the actual percentage of death in the observed population. 

### 3.5. Clinical Decision Curve Analysis (Figure 6)

DCA determines the clinical application value of nomogram model by calculating the net benefit under the probability of each death risk threshold. The abscissa of DCA is the high-risk threshold probability, and the ordinate is the net benefit (NB). When the model reaches a certain value, the probability of patient death is recorded as Pi, and it is defined as positive when Pi reaches a certain threshold (recorded as Pt). The high-risk threshold is set as (0, 1), and the net benefit rate and effective prediction probability range are evaluated by deducting the false-positive population misjudged by the model. When all patients survive or die, the nomogram model has no clinical application value. The threshold probabilities of the training set and the validation set are all between 0.01~0.94, and the net benefit rate is >0, which has clinical practical value, suggesting that the model has good clinical application value in predicting the death of severe/critical COVID-19 patients.

**Figure 6 diagnostics-12-02562-f006:**
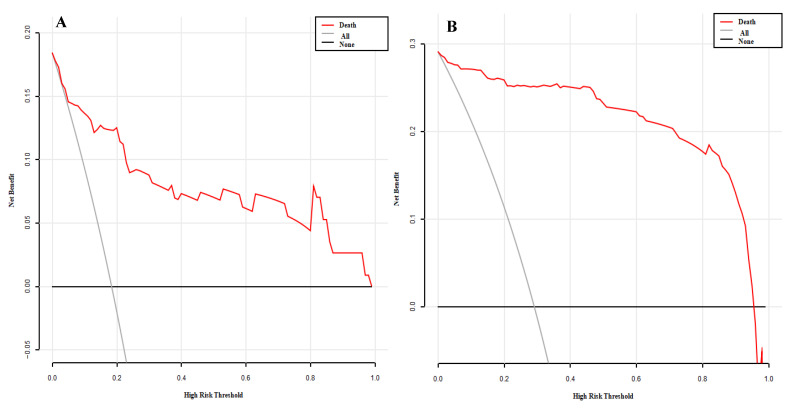
The decision curve analysis of the prediction of the mortality risk in severe/critical COVID-19 patients in the training set (**A**) and validation set (**B**).

## 4. Discussion

COVID-19 is a viral respiratory infectious disease, which has been a global pandemic in recent years and has had serious impacts on the economy and public health [11,12]. Studies have shown that patients with mild and common COVID-19 have a good prognosis, while the mortality of severe patients is 1.4~4.5% [13], and the mortality of critical patients is as high as 38.73% [14]. The prediction model is not only suitable for the allocation of limited medical resources, but also can help reduce mortality rates [15]. Compared with previously published death prediction models [4,5,7], our study mainly focused on patients with severe/critical COVID-19; we developed and externally validated a new practical model to identify patients at higher risk of death. The development and validation of the model should follow the requirements and recommendations in the Tripod statement [16]. Because the total sample size included in this study was small, with many independent variables and collinearity between different variables, Lasso regression analysis was used to screen potential independent variables. Compared with the currently commonly used stepwise regression method and optimal subset method, Lasso regression can reduce the dimension of multiple parameters and provide a solution to the collinearity problem of multiple parameters, avoiding overfitting and ensuring global optimization [17]. The model was composed of six variables, including having under three basic diseases, APACHE II score, Urea, Lac, L%, and NPR, which were obtained at admission and were all obtained from our data. The model was externally verified in the validation set, showing high discrimination and good prediction performance. Future studies may refine predictors by selecting other characteristic variables, such as CT scans, lymphocyte subtypes, organ injury markers, and cytokines.

All the predictive indicators are routine clinical testing items with fast detection speeds and low prices. Studies have found that advanced age, high APACHE II score, lymphopenia, high lactate dehydrogenase, and C-reactive protein are associated with poor prognosis [18,19]. A meta-analysis study found that the past medical histories of patients with COVID-19, including hypertension, chronic lung disease, and cardiovascular disease, may be risk factors for severe infection [20]. Our study found that certain factors, including patients with less than three previous diseases, APACHE II score, Urea, Lac, L%, and NPR, were closely related to the probability of COVID-19-related death. Different from previous studies [21], this study did not find serum amyloid A, lactate dehydrogenase, and C-reactive protein to be risk factors for death in patients with severe/critical COVID-19.

Our study found that the L% of patients who died of COVID-19 decreased significantly, indicating that the immune response might be involved in the progress of the disease. More and more evidence showed that cytokine storms play an important role in the process of disease progression to severe COVID-19 infection, and the cause of death might be related to excessive virus-induced inflammation [22]. However, we did not analyze cytokines in this study due to the lack of data. It is known that the presence of more than three basic diseases is a risk factor for COVID-19-related death, which may be related to the aggravation of inflammatory response, acute stress, and hypoxemia caused by immune dysfunction. However, the selection of immunosuppressants, such as glucocorticoids, may effectively prevent the excessive inflammation caused by viruses and selectively reduce the mortality of severe patients.

The data used in this study were those of severe patients, so this model is suitable for severely affected adult patients. Patients with low and medium death risk are suitable for the general ward, while patients with high risk of death may need to be monitored and treated in ICUs. Our model is helpful for identifying patients with high risk of death, and early intervention and appropriate treatment decisions after early identification may improve the prognosis of such patients.

However, our study still has the following limitations: First, this was a cross-regional, double-center, retrospective, observational cohort study based in China, which may not represent COVID-19 patients in other countries due to racial/ethnic differences. Therefore, international data sets and other multicenter, prospective, large-sample studies are required for external verification. Second, this study was a retrospective study, and the data obtained from the electronic medical record system were not complete; therefore, we attempted to overcome the issues raised by the missing data using the random forest method. Third, we did not collect the follow-up drug treatment of patients, and the effects of different treatment options were not taken into account in the nomogram. Fourth, some indicators reported in the literature, such as CT scan, lymphocyte subtype, organ damage markers, and cytokines, were not evaluated in our nomogram due to the lack of data. Fifth, because none of the training set patients in this study were vaccinated against COVID-19, it was not possible to determine whether vaccination affected the mortality outcome. Finally, the laboratory data may change with the progress of the disease, and it was impossible to include the dynamic changes of various indicators in the model for analysis due to the nature of retrospective studies.

## 5. Conclusions

The nomogram model presented in this study was shown to have good predictive ability and discrimination and can provide a reference for early screening of severe COVID-19 patients with high risk of death. This study carried out both internal validation and external validation in combination with the epidemic situation in Shanghai. The model has high stability, reliability, and repeatability, and is worthy of clinical promotion and application.

## Figures and Tables

**Figure 1 diagnostics-12-02562-f001:**
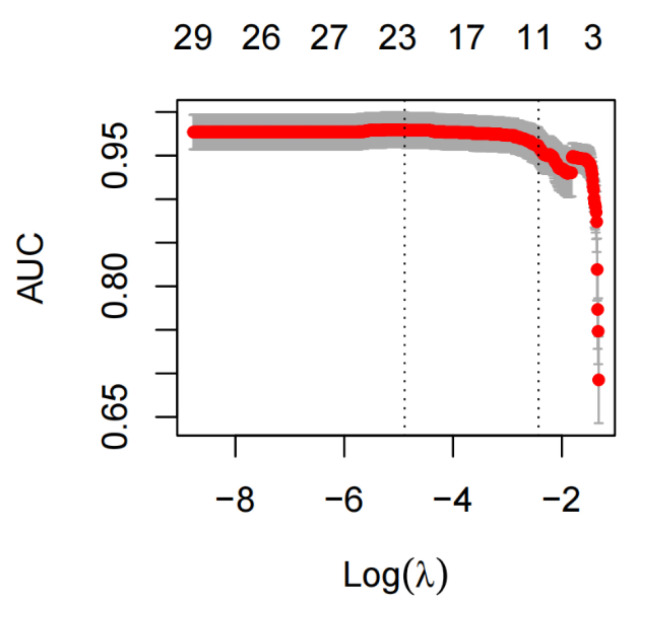
Relation curve between penalty coefficient of lasso regression variable and log lambda.

**Figure 2 diagnostics-12-02562-f002:**
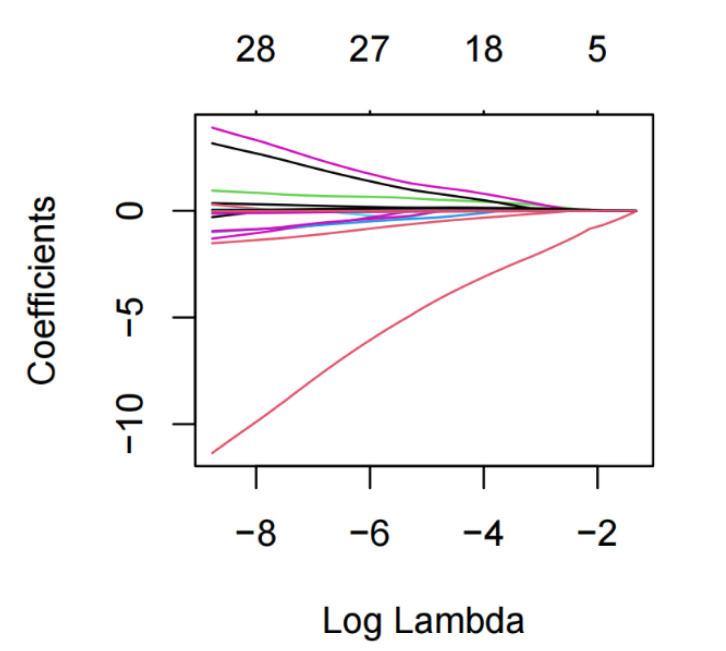
Relation curve between Lasso regression AUC and Log (λ).

**Figure 3 diagnostics-12-02562-f003:**
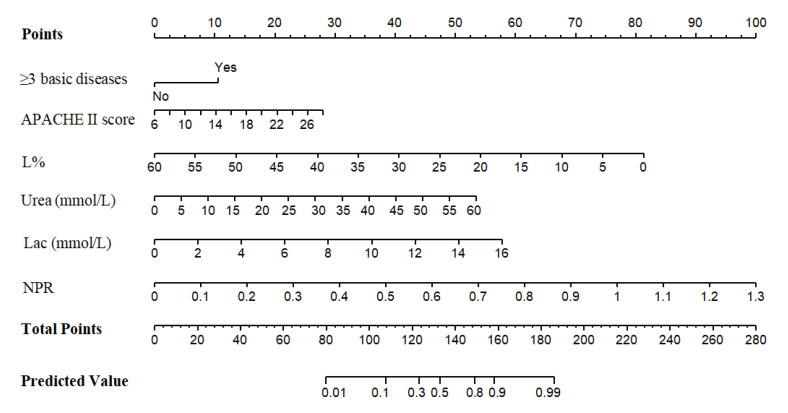
The nomogram to predict the risk of mortality in severe/critical COVID-19 patients was created based on six independent prognostic factors.

**Figure 4 diagnostics-12-02562-f004:**
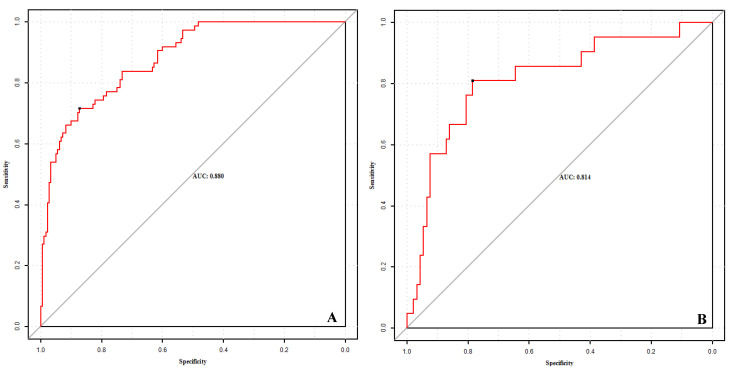
ROC for predicting the mortality among severe/critical COVID-19 patients in the training set (**A**) and validation set (**B**).

**Figure 5 diagnostics-12-02562-f005:**
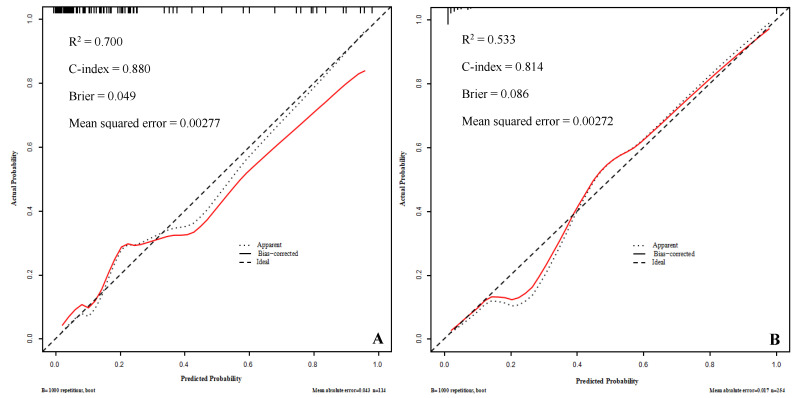
The calibration curve for the prediction of the mortality risk in severe/critical COVID-19 patients in the training set (**A**) and validation set (**B**).

**Table 1 diagnostics-12-02562-t001:** Demographic and clinical features of study population by mortality.

Characteristic	Training Set (*n* = 254)				Validation Set (*n* = 113)			
	Death Group (*n* = 74)	Survival Group (*n* = 180)	*Z*/*χ^2^* Value	*p*-Value	Death Group (*n* = 21)	Survival Group (*n* = 92)	*Z*/*χ^2^* Value	*p*-Value
Male/Female (cases)	48/26	102/78	1.458	0.227	12/9	43/49	0.714	0.389
Age [*n* (%)]	70.0 (56.8, 78.5)	62.0 (69.0, 77.0)	−4.002	<0.001	86.0 (76.5, 91.0)	82.0 (72.0, 89.0)	−1.332	0.183
18–65 Years	24 (31.6%)	88 (48.9%)	6.506	0.011	4 (19.0%)	65 (70.7%)	19.150	<0.001
≥65 Years	50 (68.4%)	92 (51.1%)			17 (81.0%)	27 (29.3%)		
COVID-19 Severe/Critical (cases)	12/62	107/73	39.354	<0.001	10/11	79/13	14.954	<0.001
Course of Disease [d, M (Q_L_, Q_U_)]	16.0 (11.8, 24.0)	28.0 (17.0, 42.0)	−4.662	<0.001	11.0 (7.0, 15.0)	8.0 (7.0, 13.5)	−0.836	0.403
Time of Nucleic Acid Turning Negative (d)	12.0 (10.0, 19.0)	13.0 (9.0, 19.0)	−0.027	0.979	6.5 (5.0, 7.3)	5.0 (5.0, 7.0)	−1.416	0.157
Clinical Manifestation [*n* (%)]								
Fever	60 (81.1%)	156 (86.7%)	1.286	0.257	14 (66.7%)	39 (42.4%)	4.046	0.044
Cough	36 (48.6%)	123 (68.3%)	8.679	0.003	15 (71.4%)	58 (63.0%)	0.526	0.468
Weakness	23 (31.1%)	77 (42.8%)	3.006	0.083	8 (38.1%)	28 (30.4%)	0.462	0.497
Diarrhea	10 (13.5%)	18 (10.0%)	0.660	0.417	1 (4.8%)	3 (3.3%)	0.000 ^a^	1.000
Dyspnea	25 (33.8%)	30 (16.7%)	9.057	0.003	4 (19.0%)	9 (9.8%)	1.442	0.230
Consciousness Disorder	5 (6.8%)	6 (3.3%)	1.483	0.223	10 (47.6%)	20 (21.7%)	5.872	0.015
Other	7 (9.5%)	15 (8.3%)	0.084	0.772	2 (9.5%)	3 (3.3%)	0.451 ^a^	0.502
Comorbidities [n (%)]								
Chronic Lung Disease	11 (14.9%)	18 (10.0%)	1.227	0.268	18 (85.7%)	46 (48.9%)	9.407	0.002
Hypertension	30 (40.5%)	63 (35.0%)	0.694	0.405	11 (52.4%)	51 (55.4%)	0.064	0.800
Diabetes	13 (24.1%)	31 (17.2%)	1.277	0.258	8 (38.1%)	26 (28.3%)	0.786	0.375
Cardiovascular Disease	11 (14.9%)	24 (13.3%)	0.104	0.748	12 (57.1%)	31 (33.7%)	3.988	0.046
Cerebrovascular Disease	7 (9.5%)	15 (8.3%)	0.084	0.772	10 (47.6%)	28 (30.4%)	2.262	0.133
Chronic Kidney Disease	12 (16.2%)	15 (8.3%)	3.430	0.064	11 (52.4%)	14 (15.2%)	13.706	<0.001
Malignancy	9 (12.2%)	24 (13.3%)	0.064	0.801	2 (9.5%)	3 (3.3%)	0.451 ^a^	0.502
≥3 Basic Diseases	14 (18.9%)	17 (9.4%)	4.393	0.036	13 (61.9%)	21 (22.8%)	12.412	<0.001
Treatment [*n* (%)]								
Mechanical Ventilation	31 (41.9%)	18 (10.0%)	34.257	<0.001	14 (66.7%)	19 (20.7%)	17.509	<0.001
ECMO	3 (4.1%)	4 (2.2)	0.151 ^a^	0698	2 (9.5%)	0 (0.0%)	-	0.033 ^b^
APACHE II score [points, M (Q_L_, Q_U_)]	17.0 (16.0, 21.0)	16.0 (15.0, 18.0)	−3.495	<0.001	20.0 (17.5, 26.0)	16.0 (12.0, 18.0)	−4.843	<0.001

Note: ECMO—extracorporeal membrane oxygenation; APACHE II score—acute physiology and chronic health evaluation II score; ^a^—the chi-square value of continuous correction; ^b^—Fisher’s test.

**Table 2 diagnostics-12-02562-t002:** Comparison of the worst laboratory indicators within 24 h for severe/critical COVID-19 patients in the training set and the validation set.

Characteristic [M (Q_L_, Q_U_)]	Training Set (*n* = 254)				Validation Set (*n* = 113)			
	Death Group (*n* = 74)	Survival Group (*n* = 180)	*Z* Value	*p*-Value	Death Group (*n* = 21)	Survival Group (*n* = 92)	*Z* Value	*p*-Value
WBC (×10^9^/L)	13.3 (9.9, 17.2)	5.7 (4.2, 7.7)	−9.407	<0.001	9.9 (6.8, 14.2)	6.1 (4.5, 10.8)	−2.160	0.031
N%	88.6 (82.3, 93.9)	73.2 (62.2, 82.6)	−7.734	<0.001	83.0 (65.2, 87.0)	79.7 (64.3, 88.2)	−0.212	0.832
N (×10^9^/L)	12,0 (8.1, 16.5)	4.3 (2.6, 6.6)	−9.496	<0.001	6.9 (4.3, 10.5)	4.7 (2.7, 9.5)	−1.725	0.084
L (×10^9^/L)	0.7 (0.4, 1.1)	0.9 (0.6, 1.2)	−3.393	0.001	0.6 (0.3, 0.9)	0.8 (0.5, 1.4)	−0.793	0.428
L%	5.1 (2.9, 7.7)	12.7 (8.4, 18.6)	−8.802	<0.001	12.0 (6.9, 21.6)	14.1 (9.7, 20.6)	−0.793	0.428
NLR	18.0 (9.3, 34.1)	4.3 (2.4, 8.0)	−9.115	<0.001	10.0 (5.0, 20.9)	5.4 (2.0, 15.8)	−2.361	0.018
Hb (g/L)	108.0 (89.0, 127.3)	124.0 (111.0, 135.0)	−4.273	<0.001	99.0 (86.0, 125.0)	121.0 (102.0, 136.0)	−2.000	0.046
PLT (×10^9^/L)	131.5 (64.3, 203.0)	191.0 (143.0, 257.0)	−4.648	<0.001	115.0 (79.5, 187.0)	183.0 (133.0, 231.0)	−2.592	0.010
NPR	0.2 (0.1, 0.4)	0.02 (0.01, 0.03)	−10.018	<0.001	0.05 (0.03, 0.13)	0.03 (0.02, 0.05)	−3.088	0.002
CRP (mg/L)	31.4 (11.8, 70.6)	12.6 (0.5, 61.9)	−2.612	0.009	95.0 (27.5, 171.0)	25.4 (7.1, 88.0)	−2.394	0.017
SAA (mg/L)	134.7 (63.6, 200.0)	47.7 (11.5, 126.9)	−4.426	<0.001	146.2 (52.7, 248.1)	82.0 (10.4, 225.1)	−1.601	0.109
PCT (μg/L)	2.3 (0.8, 5.6)	0.1 (0.04, 0.19)	−10.414	<0.001	1.6 (0.3, 8.0)	0.5 (0.4, 1.0)	−4.493	<0.001
ALT (U/L)	51.0 (21.8, 121.0)	30.0 (21.0, 51.0)	−3.583	<0.001	26.0 (14.0, 40.5)	19.0 (14.5, 29.5)	−1.174	0.240
AST (U/L)	81.0 (40.0, 159.8)	33.0 (23.0, 45.8)	−7.115	<0.001	41.0 (23.5, 52.5)	30.0 (20.5, 49.5)	−1.324	0.185
TBIL (μmol/L)	21.0 (16.0, 37.1)	10.4 (8.4, 15.6)	−7.012	<0.001	14.7 (11.5, 16.9)	11.2 (9.3, 16.6)	−1.590	0.112
DBIL (μmol/L)	10.7 (7.4, 21.0)	4.0 (3.0, 5.8)	−8.266	<0.001	5.6 (3.2, 7.8)	2.8 (2.2, 3.9)	−3.573	<0.001
ALB (g/L)	30.2 (27.9, 33.2)	36.0 (32.8, 38.4)	−7.728	<0.001	30.0 (28.5, 33.5)	37.0 (30.0, 39.0)	−3.094	0.002
Urea (mmol/L)	16.7 (9.1, 31.2)	4.9 (3.9, 7.3)	−9.112	<0.001	12.3 (7.7, 21.0)	8.3 (5.6, 11.7)	−3.023	0.003
Cr (μmol/L)	121.5 (62.8, 337.8)	61.0 (51.0, 78.0)	−5.987	<0.001	102.0 (69.5, 137.0)	77.0 (63.0, 105.5)	−2.632	0.008
eGFR (ml/min)	51.7 (15.3, 93.2)	98.9 (85.9, 107.4)	−7.383	<0.001	56.0 (39.5, 80.0)	80.0 (54.0, 91.5)	−2.358	0.018
Lac (mmol/L)	2.9 (2.1, 4.2)	1.9 (1.4, 2.5)	−6.942	<0.001	2.6 (2.1, 4.1)	1.5 (1.1, 2.0)	−4.442	<0.001
BNP (ng/L)	1619.0 (392.1, 2196.5)	326.9 (133.7, 682.2)	−6.314	<0.001	638.0 (197.0, 899.0)	115.0 (45.5, 238.0)	−4.543	<0.001
cTnI (gl/L)	0.7 (0.1, 3.5)	0.1 (0.1, 0.3)	−3.647	<0.001	0.2 (0.1, 0.3)	0.1 (0.1, 0.2)	−3.583	<0.001
PT (s)	12.4 (12.6, 16.8)	13.0 (11.8, 83.8)	−0.132	0.895	12.9 (12.3, 13.3)	11.7 (11.0, 12.4)	−3.797	<0.001
APTT (s)	32.0 (26.2, 36.3)	28.9 (26.2, 32.6)	−2.923	0.003	33.7 (27.3, 36.3)	30.6 (28.2, 34.8)	−0.716	0.474
D-dimer (mg/L)	6.4 (2.3, 22.6)	1.4 (0.6, 4.1)	−6.524	<0.001	4.5 (1.7, 8.4)	1.1 (0.5, 3.4)	−3.827	<0.001

Note: WBC—white blood cell; N%—percentage of neutrophils; N—neutrophils; L—lymphocytes; L%—percentage of lymphocytes; NLR—neutrophil-to-lymphocyte ratio; Hb—hemoglobin; PLT—platelet; NPR—neutrophil-to-platelets ratio; CRP—C-reactive protein; SAA—serum amyloid A; PCT—procalcitonin; ALT—alanine aminotransferase; AST—aspartate aminotransferase; TBIL—total bilirubin; DBIL—direct bilirubin; ALB—albumin; Urea—urea nitrogen; Cr—creatinine; eGFR—estimated glomerular filtration rate; Lac—lactic acid; BNP—B-type natriuretic peptide; cTnI—troponin; PT—prothrombin time; APTT—activated partial thrombin time.

**Table 3 diagnostics-12-02562-t003:** Multifactor logistics regression analysis of mortality of severe/critical COVID-19 patients.

Variable	β	SE	Wald	*p*	OR	95%CI
Age ≥ 65	1.424	0.74	3.697	0.054	4.153	0.973~17.728
≥3 Basic Diseases	1.732	0.757	5.242	0.022	15.653	1.283~24.904
APACHE II Score	0.190	0.095	4.012	0.045	1.209	1.004~1.456
L%	−0.253	0.079	10.253	0.001	0.777	0.665~0.907
Urea	0.177	0.060	8.715	0.003	1.193	1.061~1.342
D-dimer	0.002	0.019	0.009	0.925	1.002	0.966~1.039
Lac	0.750	0.221	11.491	0.001	2.117	1.372~3.266
NPR	11.333	4.960	5.220	0.022	8.056	5.008~13.819
NLR	−0.001	0.007	0.048	0.826	0.999	0.985~1.012
WBC	0.032	0.074	0.183	0.669	1.032	0.893~1.193
cTnI	−0.029	0.013	4.961	0.056	0.971	0.947~0.997
Constant	−8.309	1.989	17.448	0.000	0.000	--

## Data Availability

The original contributions presented in the study are included in the article, and further inquiries can be directed to the corresponding author.

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
