# Peer review of "Construction and Validation of Mortality Risk Nomograph Model for Severe/Critical Patients with COVID-19"

_diagnostics, 2022, doi:10.3390/diagnostics12102562_

Round 1

Reviewer 1 Report

The manuscript describes a detailed assessment, development and validation of a model of mortality risk in COVID-19 infections for adult patients.

Specific points to address:

1) Page 3, paragraph 1 (93-98). Please rephrase this paragraph as it is very difficult to read and understand.

2) Page 3 (lines 116 -118). 'Nucleic acid turned negative'. Please explain what this is/means? If covid testing is undertaken for patients, please make this clear in materials and methods, i.e were lateral flow tests and/or PCR based methodology for confirming COVID infection status?

3) There is no mention within the data collection as to whether patients were vaccinated or not? Was this information collected or not available?

4) Whilst the authors acknowledge that the model they describe may only be of use in a specfic geographical/population based area, they have not discussed the effect of vaccination or new variants of SARS COVID 19 such as Omicron BA.4 and BA.5 and how these may drastically alter their nomograph model of mortality risk. This should be addressed in the discussion.

5) Have the Authors developed an 'App' based on these findings that could be evaluated by medical staff in the hospital? This would seem the logical outcome of the study particularly in terms of guiding clinical decision making and also allow prospective collection of data to support their findings.

Author Response

Point 1: Page 3, paragraph 1 (93-98). Please rephrase this paragraph as it is very difficult to read and understand.

Response 1: I am very sorry for your misunderstanding due to my unprofessional English writing. I have revised it according to professional English writing and this paragraph has been revised as follows: The general information, physical examination and auxiliary examination data of the patients were collected respectively, and the classification was determined according to the recommendations of the guidelines [1-2]. The worst blood samples of patients with severe/critical COVID-19 within 24 hours after admission were collected, including blood routine, blood biochemistry, coagulation function, blood gas analysis and other related indexes, and the scores of acute physiology and chronic health evaluation â…¡ (APACHE II) were calculated.

Point 2: Page 3 (lines 116 -118). 'Nucleic acid turned negative'. Please explain what this is/means? If covid testing is undertaken for patients, please make this clear in materials and methods, i.e were lateral flow tests and/or PCR based methodology for confirming COVID infection status?

Response 2: Thank you very much for your professional review. In view of my unclear presentation, page 3 (lines 122 - 130) has been revised as follows: 74 (29.13%) of 254 patients in the training set eventually died, of whom 57 (22.44%) patients were still positive at the time of death, but 17 (6.69%) patients were negative by two consecutive coronavirus nucleic acid tests before death (sampling time at least 24 hours apart). The causes of death were other irreversible serious complications or basic diseases. 21 (18.58%) of the 113 patients in the validation set eventually died, of whom 5 (4.42%) patients were still positive by the novel coronavirus nucleic acid test at the time of death, and 16 (14.16%) patients were negative by two consecutive novel coronavirus nucleic acid tests before death (at least 24 h apart at the time of sampling) and the causes of death were other irreversible serious complications or basic diseases.

Thank you very much for your professional advice. The third inclusion criteria has been added to Page 2 (lines 71-72), that is, patients were positive for the novel coronavirus nucleic acid detected by fluorescence quantitative PCR.

Point 3:There is no mention within the data collection as to whether patients were vaccinated or not? Was this information collected or not available?

Response 3: Thank you very much for your professional advice. The reason why COVID-19 vaccination is not mentioned in the data is that none of the patients in Wuhan in 2020 were vaccinated against COVID-19 at that time. The vaccination situation of Shanghai patients was collected in 2022. Among the 113 patients, 26 (22.01%) received 3 doses of COVID-19 vaccine, 32 (28.32%) received 2 doses of COVID-19 vaccine, 41 (36.28%) received 1 dose of COVID-19 vaccine, and 14 (12.39%) did not receive COVID-19 vaccine. Of the 21 patients in the death group, 1 case (4.76%) received 3 doses of COVID-19 vaccine, 4 cases (19.05%) received 2 doses of COVID-19 vaccine, 7 cases (33.33%) received 1 dose of COVID-19 vaccine, and 8 cases (38.10%) did not receive COVID-19 vaccine. Of the 92 patients in the survival group, 25 cases (27.17%) received 3 doses of COVID-19 vaccine, 28 cases (30.43%) received 2 doses of COVID-19 vaccine, 34 cases (36.96%) received 1 dose of COVID-19 vaccine, and 6 cases (6.52%) did not receive COVID-19 vaccine. However, because there is no comparative analysis of Wuhan patients, this variable is not suitable for this study.

Point 4:Whilst the authors acknowledge that the model they describe may only be of use in a specfic geographical/population based area, they have not discussed the effect of vaccination or new variants of SARS COVID 19 such as Omicron BA.4 and BA.5 and how these may drastically alter their nomograph model of mortality risk. This should be addressed in the discussion.

Response 4: Thank you very much for your rigorous review. Based on your consideration, I once again conducted statistical analysis of the verification set data in Shanghai in 2022 and found that there was a difference in the vaccination rate of COVID-19 vaccine between the survival group and the death group (c2 =14.117, P<0.001). This is consistent with the conclusion that COVID-19 vaccination can effectively reduce the case fatality rate reported in the literature [3-5]. Then the variable of whether or not to be vaccinated against COVID-19 was analyzed by univariate Logistic regression analysis [P<0.001, 95%CI(0.034--0.380)], and then combined with the 6 characteristic variables in this study to conduct multivariate Logistic regression analysis [P=0.027, 95%CI(0.032--0.814)]. According to the above analysis, the variable of vaccination or not may change the construction of nomograph for mortality risk, which is also one of the important reference variables for us to continue to develop the nomograph model in the future. However, the Wuhan patients in 2020 in this study were not vaccinated with COVID-19 vaccine at that time and could not become a candidate variable in the training set, so it was not suitable for this study.

The main epidemic strain of Wuhan in 2020 is Delta, while the main epidemic strain of Shanghai in 2022 is Omicron BA.2. In this study, there was a significant difference in the mortality between the training set and the verification set (c2 =4.537,P=0.033). The possible reasons are as follows. First of all, COVID-19 vaccination can effectively reduce mortality [3-4]. Secondly, although the transmission ability of Omicron strain is stronger than that of Delta strain, its pathogenicity is weakened [6]. Finally, early and aggressive administration of Paxlovid can significantly reduce the PCR conversion time [7]. A recent study has found that Paxlovid can increase the negative conversion rate of COVID-19 nucleic acid [8].

 Some studies have shown that Omicron sublineages BA.2.12.1, BA.4 and BA.5 exhibit higher transmissibility than the BA.2 lineage [9], but the related mortality reports are very few, which is also the direction that we need to further study in the future.

Point 5:Have the Authors developed an 'App' based on these findings that could be evaluated by medical staff in the hospital? This would seem the logical outcome of the study particularly in terms of guiding clinical decision making and also allow prospective collection of data to support their findings.

Response 5:Thank you very much for your professional review. At present, our team is cooperating with bioengineers and computer engineers to develop a comprehensive clinical information database and artificial intelligence diagnosis system for COVID-19. A comprehensive database is constructed by large sample, multi-center case collection and integration of clinical diagnosis and treatment of COVID-19 patients big data, and the artificial intelligence diagnosis software "COVID-19 unbiased prediction based on mixed learning" is designed to realize the efficient fusion of clinical diagnosis and treatment data. The system can accurately predict the potential death risk of COVID-19 patients. It provides important reference information for clinicians to make decisions by performing clinical data integration and building a model for COVID-19 patients.

References:

  • Zhenqiang BI, Qingwu J, Peng W, et al. Guidance for COVID-19 prevention, control, diagnosis and management. People’s Medical Publishing House. First published: 2020. ISBN 978-7-117-29817-9.
  • General Office of the National Health Commission.Notice on printing and distributing the diagnosis and treatment plan for COVID-19 (trial version 9) [EB/OL]. State Health Commission of the people's Republic of China (2022-03-14) [2022-03-14]. http://www.nhc.gov.cn/yzygj/s7653p/202203/b74ade1ba4494583805a3d2e40093d88.shtml.
  • Lopez Bernal J, Andrews N, Gower C, et al. Effectiveness of the Pfizer-BioNTech and Oxford-AstraZeneca vaccines on covid-19 related symptoms, hospital admissions, and mortality in older adults in England: test negative case-control study[J]. BMJ, 2021, 373: n1088. doi: 10.1136/bmj.n1088.
  • Sadoff J, Gray G, Vandebosch A, et al. Safety and Efficacy of Single-Dose Ad26.COV2.S Vaccine against Covid-19[J]. N Engl J Med, 2021,384(23):2187-2201. doi: 10.1056/NEJMoa2101544.
  • Hodgson SH, Mansatta K, Mallett G, et al. What defines an efficacious COVID-19 vaccine? A review of the challenges assessing the clinical efficacy of vaccines against SARS-CoV-2[J]. Lancet Infect Dis,2021,21(2):e26-e35. doi: 10.1016/S1473-3099(20)30773-8.
  • Papanikolaou V, Chrysovergis A, Ragos V, et al. From delta to Omicron: S1-RBD/S2 mutation/deletion equilibrium in SARS-CoV-2 defined variants[J]. Gene, 2022,814:146134. doi: 10.1016/j.gene.2021.146134.
  • JS Shao, R Fan, JR Hu, et al. Clinical Progression and Outcome of Hospitalized Patients Infected with SARS-CoV-2 Omicron Variant in Shanghai, China[J].Vaccines (Basel), 2022,10(9):1409. doi: 10.3390/vaccines10091409.
  • Zhong W, Jiang X, Yang X, et al. The efficacy of paxlovid in elderly patients infected with SARS-CoV-2 omicron variants: Results of a non-randomized clinical trial[J].Front Med (Lausanne), 2022,9:980002. doi: 10.3389/fmed.2022.980002.
  • Cao Y, Yisimayi A, Jian F, et al. BA.2.12.1, BA.4 and BA.5 escape antibodies elicited by Omicron infection[J].Nature,2022,608(7923):593-602. doi: 10.1038/s41586-022-04980-y.

Reviewer 2 Report

The authors presented a good manuscript presenting a brand new algorithm for the evaluation of prognosis in COVID-19 patients. The aims as well as tasks of the study are clear. Methods seem to be adequate and well described. Apparently, established algorithm may serve a useful tool in clinical practice. Moreover, the authors revealed a valuable information regarding the important signs of the high risk of severe course of the disease.

I have only two edits to propose before the final acceptance:

1) The authors more than once indicate that the novel algorithm may help to reveal the patients with the high risk of severe course of COVID-19 and therefore may decrease the mortality. The only question is how this information about threatened patients can be bridged with increase of survival level? In other words, how the knowledge that the person has the increased risk may influence the decision-making in terms of medical interventions? Please, describe your view more detailed.

2) I suppose that it is not very well to use the forms like "don't" rather than "do not" in scientific texts. Kindly avoid any types of constriction when preparing the articles.

Author Response

Point 1: The authors more than once indicate that the novel algorithm may help to reveal the patients with the high risk of severe course of COVID-19 and therefore may decrease the mortality. The only question is how this information about threatened patients can be bridged with increase of survival level? In other words, how the knowledge that the person has the increased risk may influence the decision-making in terms of medical interventions? Please, describe your view more detailed.

Response 1:Thank you very much for your review of the article and professional suggestions. For patients detected to be at high risk of mortality after risk screening, we should do as follows. First of all, as critical care physicians, we would pay enough attention to optimize medical resources. Secondly, we will actively start the expert consultation of COVID-19 MDT. Finally, we will perform preemptive therapy, such as early antiviral, immunotherapy, early and effective respiratory support and organ protection.

Point 2: I suppose that it is not very well to use the forms like "don't" rather than "do not" in scientific texts. Kindly avoid any types of constriction when preparing the articles.

Response 2:Thank you very much for your preciseness and carefulness. All "don't" in the full text have been replaced with "do not".

Round 2

Reviewer 1 Report

Thank you for addressing the issues I raised in my original review. The changes you have made have significantly improved the content of the manuscript.